# C9orf72-derived arginine-rich poly-dipeptides impede phase modifiers

Hitoki Nanaura [1,2,19], Honoka Kawamukai[3,4,19], Ayano Fujiwara[5,19], Takeru Uehara[5], Yuichiro Aiba [6], Mari Nakanishi[2], Tomo Shiota[1,2], Masaki Hibino[6], Pattama Wiriyasermkul[7,18], Sotaro Kikuchi[2], Riko Nagata [2], Masaya Matsubayashi [2], Yoichi Shinkai [8], Tatsuya Niwa[9], Taro Mannen [5], Naritaka Morikawa [2], Naohiko Iguchi[1], Takao Kiriyama [1], Ken Morishima[10], Rintaro Inoue[10], Masaaki Sugiyama[10], Takashi Oda[11,12], Noriyuki Kodera[13], Sachiko Toma-Fukai[14], Mamoru Sato[11], Hideki Taguchi [9], Shushi Nagamori [7,18], Osami Shoji [6], Koichiro Ishimori [3,15], Hiroyoshi Matsumura [5], Kazuma Sugie[1], Tomohide Saio [4,16✉], Takuya Yoshizawa[5✉] & Eiichiro Mori [2,17✉]

Nuclear import receptors (NIRs) not only transport RNA-binding proteins (RBPs) but also modify phase transitions of RBPs by recognizing nuclear localization signals (NLSs). Toxic arginine-rich poly-dipeptides from *C9orf72* interact with NIRs and cause nucleocytoplasmic transport deficit. However, the molecular basis for the toxicity of arginine-rich poly-dipeptides toward NIRs function as phase modifiers of RBPs remains unidentified. Here we show that arginine-rich poly-dipeptides impede the ability of NIRs to modify phase transitions of RBPs. Isothermal titration calorimetry and size-exclusion chromatography revealed that proline:arginine (PR) poly-dipeptides tightly bind karyopherin-β2 (Kapβ2) at 1:1 ratio. The nuclear magnetic resonances of Kapβ2 perturbed by PR poly-dipeptides partially overlapped with those perturbed by the designed NLS peptide, suggesting that PR poly-dipeptides target the NLS binding site of Kapβ2. The findings offer mechanistic insights into how phase transitions of RBPs are disabled in *C9orf72*-related neurodegeneration.

[1] Department of Neurology, Nara Medical University, Kashihara, Nara, Japan. [2] Department of Future Basic Medicine, Nara Medical University, Kashihara, Nara, Japan. [3] Graduate School of Chemical Sciences and Engineering, Hokkaido University, Sapporo, Hokkaido, Japan. [4] Graduate School of Medical Sciences, Tokushima University, Tokushima, Japan. [5] College of Life Sciences, Ritsumeikan University, Kusatsu, Shiga, Japan. [6] Department of Chemistry, Graduate School of Science, Nagoya University, Nagoya, Aichi, Japan. [7] Laboratory of Bio-Molecular Dynamics, Department of Collaborative Research, Nara Medical University, Kashihara, Nara, Japan. [8] Molecular Neurobiology Research Group, Biomedical Research Institute, National Institute of Advanced Industrial Science and Technology (AIST), Tsukuba, Ibaraki, Japan. [9] Cell Biology Center, Institute of Innovative Research, Tokyo Institute of Technology, Yokohama, Japan. [10] Institute for Integrated Radiation and Nuclear Science, Kyoto University, Kyoto, Japan. [11] Graduate School of Medical Life Science, Yokohama City University, Yokohama, Japan. [12] Department of Life Science, Rikkyo University, Toshima ku, Tokyo, Japan. [13] Nano Life Science Institute (WPI-NanoLSI), Kanazawa University, Kanazawa, Japan. [14] Division of Materials Science, Graduate School of Science and Technology, Nara Institute of Science and Technology, Ikoma, Nara, Japan. [15] Department of Chemistry, Faculty of Science, Hokkaido University, Sapporo, Hokkaido, Japan. [16] Institute of Advanced Medical Sciences, Tokushima University, Tokushima, Japan. [17] V-iCliniX Laboratory, Nara Medical University, Kashihara, Nara, Japan. [18] Present address: Department of Laboratory Medicine, The Jikei University School of Medicine, Tokyo, Japan. [19] These authors contributed equally: Hitoki Nanaura, Honoka Kawamukai, Ayano Fujiwara. ✉email: saio@tokushima-u.ac.jp; t-yosh@fc.ritsumei.ac.jp; emori@naramed-u.ac.jp

Low-complexity protein sequences (LC-domains), regions with little diversity in the amino acid composition, are often found in RNA-binding proteins (RBPs) and are prone to self-associate, to drive phase transitions into liquid-like or gel-like states[1–4]. Regulating self-association of RBPs, including fused in sarcoma (FUS), TAR DNA-binding protein of 43 kDa (TDP43), and other heterogeneous nuclear ribonucleoproteins (hnRNPs), to suppress the formation of pathogenic fibrils, is crucial in the prevention of neurodegenerative diseases[5,6]. Nuclear import receptor (NIR) karyopherin-β2 (Kapβ2) not only controls the nucleocytoplasmic distribution of RBPs but also acts as a phase modifier to regulate self-association of FUS by recognizing proline–tyrosine nuclear localization signal (NLS)[7–10]. NIRs Importinα/Importinβ1 complex (Impα/β1) also modify self-association of TDP43[8]. Decreased affinity of Kapβ2 and FUS leads to aberrant cytoplasmic localization of FUS[11], which is observed in amyotrophic lateral sclerosis (ALS) pathology[12,13].

A hexanucleotide repeat expansion in *C9orf72* is the most prevalent form of familial ALS and frontotemporal dementia (C9-ALS/FTD)[14]. Similar to other forms of ALS, mislocalization and aggregation of RBPs are observed in C9-ALS/FTD[12,13]. The main mechanisms of C9-ALS/FTD have been proposed as follows: *C9orf72* haploinsufficiency, toxic gain-of-function from repeat RNA or poly-dipeptides, or some combination of the above[15]. Repeat expansion in *C9orf72* produces five different poly-dipeptides, proline:arginine (PR), glycine:arginine (GR), proline:alanine (PA), glycine:proline (GP), and glycine:alanine (GA), through repeat-associated non-AUG (RAN) translation[16,17].

Arginine-rich, PR and GR, poly-dipeptides show similarly high toxic effect on membraneless organelles, proteins with LC-domains, and nucleocytoplasmic transport (NCT)[14,18,19]. Arginine-rich poly-dipeptides bind proteins with LC-domains and stabilize polymers formed by protein–protein interaction through LC-domains[2,4]. Existing genetic studies reveal that the repeat expansion in *C9orf72* gene disrupts NCT[20–22]. One mechanism of the disruption of NCT is that PR poly-dipeptides bind and stabilize phenylalanine:glycine-rich domains of nuclear pore proteins[3], although controversial results have also been reported[23].

Another proposed mechanism of the impairment of NCT is the toxicity of arginine-rich poly-dipeptides towards NIRs. Proteomic studies have indicated that NIRs are potential interactors of arginine-rich poly-dipeptides[2,4]. More recently, arginine-rich poly-dipeptides have been found to interact with NIRs and cause NCT deficit[24,25], although the molecular basis for the way in which arginine-rich poly-dipeptides affect NIRs function as phase modifiers remains elusive.

In this study, we investigated the effect of arginine-rich poly-dipeptides on phase modifiers through the use of multiple biochemical and biophysical methods. The approaches we utilized in this study include the following: (i) an assessment of phase transitions by droplet formation and hydrogel binding assay, (ii) an interaction analysis in a cell and a test tube, and (iii) a detailed solution nuclear magnetic resonance (NMR) analysis verified by molecular dynamics (MD) simulation. We describe here how arginine-rich poly-dipeptides impede phase modifiers, providing mechanistic insights into *C9orf72*-related neurodegeneration.

## Results

**Arginine-rich poly-dipeptides impede NIRs function of modifying RBP phase transitions**. First, we examined the effect of five different poly-dipeptides on Kapβ2 in terms of melting full-length FUS droplets (Fig. 1a, b). Turbidity assessment showed that Kapβ2 kept FUS from changing into liquid-like droplets and the addition of equimolar of PR/GR poly-dipeptides inhibited the

ability of Kapβ2, whereas that of PA/GP/GA poly-dipeptides did not (Fig. 1c). This suggests that arginine-rich poly-dipeptides disable the Kapβ2 function of melting FUS droplets. To further investigate, we focused on PR poly-dipeptides and evaluated their effect on Kapβ2 activity. We observed with fluorescence microscopy that FUS forms liquid-like droplets and these FUS droplets are dissolved by Kapβ2 (Fig. 1d). By contrast, Kapβ2 loses the ability to suppress the phase transition of FUS in the presence of PR poly-dipeptides (Fig. 1d). As previously reported[7], the designed NLS peptide—M9M, an inhibitor for Kapβ2—abolished the suppression of the phase transition of FUS (Fig. 1d and Supplementary Fig. 1a). These data suggest that PR poly-dipeptides disable the Kapβ2 function of dissolving FUS droplets.

Next, we performed hydrogel binding assay to test the effect of PR poly-dipeptides on the ability of Kapβ2 in preventing polymerization of the LC-domain (Fig. 1e). Hydrogels of mCherry fusion LC-domain of FUS (mCh:FUS-LC) were incubated with green fluorescent protein (GFP) fusion FUS-LC (GFP:FUS-LC) or GFP:FUS-LC fusion FUS-NLS (501–526, GFP:FUS-LC:NLS; Fig. 1a). Hydrogel binding of GFP:FUS-LC was not blocked (Supplementary Fig. 1b), whereas that of GFP:FUS-LC:NLS was blocked by Kapβ2 (Fig. 1f, g), suggesting that Kapβ2 recognizes NLS of FUS and keeps FUS-LC monomeric. In addition, we observed that PR poly-dipeptides inhibited the ability of Kapβ2 from blocking the hydrogel binding of GFP:FUS-LC:NLS (Fig. 1h, i) and hnRNPA2-LC, another RBP interacting with Kapβ2 (Supplementary Fig. 1c, d), suggesting that PR poly-dipeptides impede Kapβ2 function of regulating RBPs phase transitions. We also observed that PR poly-dipeptides inhibited the ability of other NIRs Impα/β1 in modifying the phase transition of LC-domain of TDP43 (TDP43-LC) (Supplementary Fig. 1e, f). Together, these results indicate that PR poly-dipeptides impede NIRs from modifying RBPs phase transitions.

**PR poly-dipeptides directly bind Kapβ2**. To confirm the interaction between Kapβ2 and PR poly-dipeptides in a cellular environment, we performed immunoprecipitation (Fig. 2a and Supplementary Fig. 2a, b). We observed that endogenous Kapβ2 was co-immunoprecipitated with PR poly-dipeptides expressed in HeLa cells (Fig. 2a). Next, we performed on-column binding assay with purified recombinant proteins, to examine the direct interaction between PR poly-dipeptides and Kapβ2. We found that both MBP:PR18 (18 repeats of Pro-Arg) and Glutathione S-transferase (GST):PR18 bound Kapβ2 (Fig. 2b and Supplementary Fig. 2c), whereas MBP:PR8 (8 repeats of Pro-Arg) did not (Fig. 2b). We also found that GST:PR18 did not bind to control proteins with variety pI values (Supplementary Fig. 2d). We observed interactions between MBP:PR18 with other NIRs Impβ1 or its adapter protein Impα by on-column binding assay (Supplementary Fig. 2e). These results suggest that PR poly-dipeptides target NIR family proteins. In addition, we used isothermal titration calorimetry (ITC) and size-exclusion chromatography with multi-angle light scattering (SEC-MALS) to analyze the interaction quantitatively. ITC revealed that MBP:PR18 bound to Kapβ2 at a $K_d$ value of 81.3 nM (Fig. 2c). MBP:PR18 bound Kapβ2 more strongly than full-length FUS does (Supplementary Table 1). SEC-MALS showed that MBP:PR18 bound Kapβ2 in a 1 : 1 ratio (Fig. 2d). We also confirmed the complex formation of MBP:PR18 and Kapβ2 by analytical ultracentrifugation (AUC) (Supplementary Fig. 2f). These results indicate that repeat extended PR poly-dipeptides directly interact with Kapβ2.

**PR poly-dipeptides target the NLS-binding site of Kapβ2**. We further investigated the binding sites of PR poly-dipeptides on

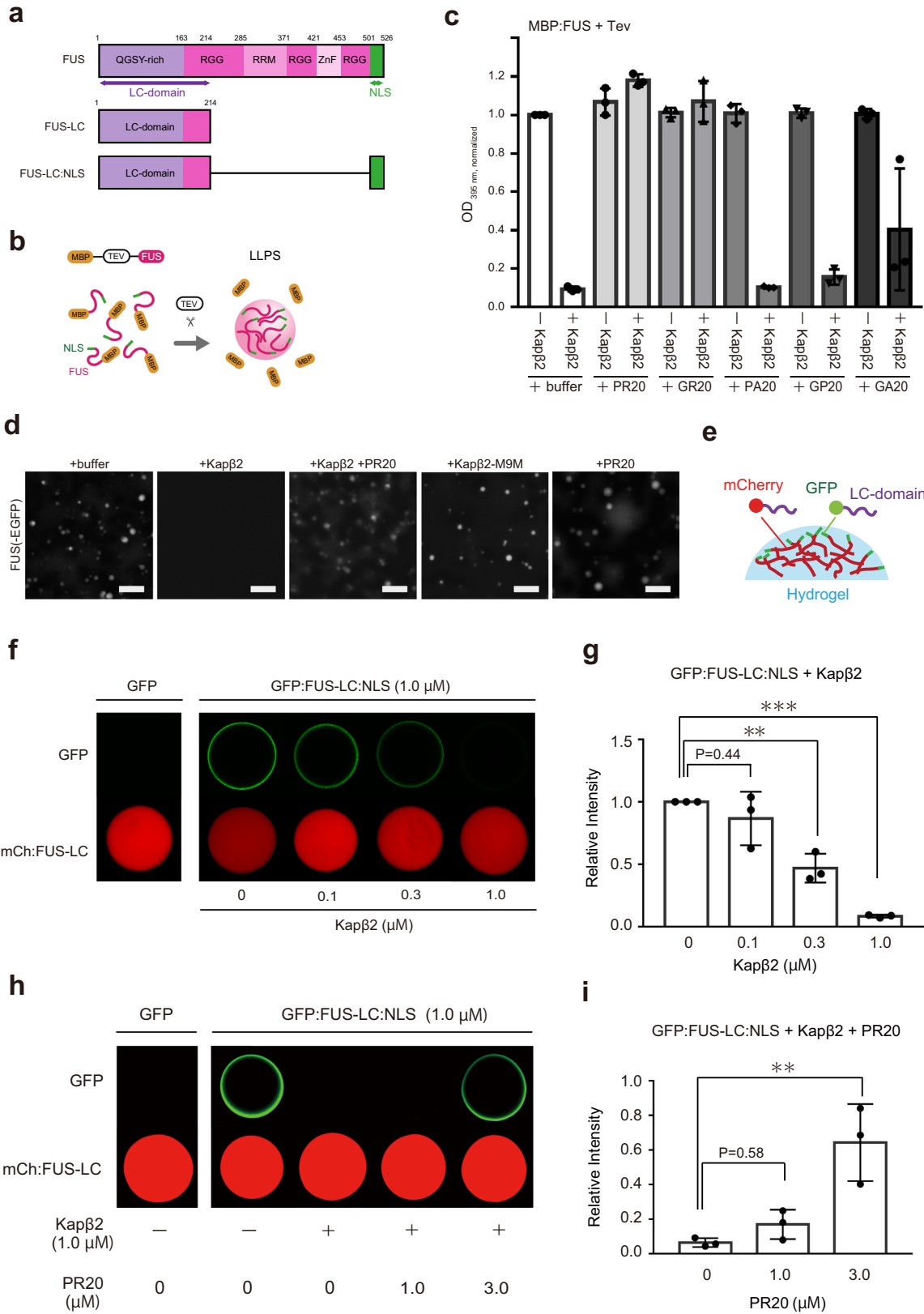

Kapβ2 using solution NMR. Despite the large size of Kapβ2 (100 kDa)[11], the use of advanced NMR techniques, including methyl-selective isotope labeling (Fig. 3a) and methyl-transverse relaxation-optimized spectroscopy[26,27], achieved high-quality NMR spectra (Fig. 3b and Supplementary Fig. 3). NMR spectra of the isotopically labeled Kapβ2 were acquired in the absence

and presence of PR20 (Fig. 3c). The addition of PR20 induced significant perturbations to several resonances of Kapβ2 (Fig. 3c), indicating that PR poly-dipeptides bind to specific regions of Kapβ2.

In order to obtain information about the binding site of PR poly-dipeptides, we performed reference NMR experiments. The

**Fig. 1 Arginine-rich poly-dipeptides impede the Kapβ2 function of modifying FUS phase transitions. a** Domain architecture of FUS, LC-domain of FUS (FUS-LC), and FUS-LC fusion NLS (FUS-LC:NLS). **b** Graphical representation of FUS droplet formation for microscope observation and turbidity assay. **c** Turbidity of 8 μM MBP:FUS in the presence of buffer, ±8 μM Kapβ2, and ±8 μM PR20/GR20/PA20/GP20/GA20. OD 395 nm is normalized to measurement of MBP:FUS + buffer + Tev. Mean of three technical replicates, ±SD. **d** Microscopic images of FUS droplets in the absence and presence of Kapβ2 and/or PR20 show that FUS droplets were dissolved by Kapβ2 and not melted in the presence of PR20. Mixture of 7.6 μM MBP:FUS and 0.4 μM MBP:FUS:EGFP were treated with TEV for an hour in the presence or absence of 16 μM Kapβ2, 16 μM Kapβ2–M9M complex, and 50 μM PR20. The experiment was independently repeated three times with similar results. Here, 10 μm scale bars are shown in the image. **e** Graphical description of hydrogel binding assay. Hydrogels of mCh:LC-domain accumulate GFP:LC-domain as it co-polymerizes. **f** Hydrogel binding assay for FUS-LC in the absence and presence of Kapβ2. mCh:FUS-LC (lower images) were incubated with 1.0 μM of GFP (left panel) or 1.0 μM of GFP:FUS-LC:NLS (right panel) in the presence of different concentrations of Kapβ2 (left to right: 0.1, 0.3, and 1.0 μM). **g** Quantitative analysis of Fig. 1f. Relative intensity of GFP signals is shown as the mean of three independent experiments ± SD, analyzed by one-way ANOVA followed by Dunnett's multiple comparison test (**$P < 0.01$, ***$P < 0.001$). **h** Hydrogel binding assay for FUS-LC in the absence and presence of Kapβ2 and/or PR20. Hydrogel droplets of mCh:FUS-LC (lower images) were incubated with 1.0 μM of GFP (left panel) or GFP:FUS-LC:NLS (right panel). GFP:FUS-LC:NLS containing Kapβ2 (1.0 μM) was challenged for homotypic polymer extension in the absence or presence of different concentration of PR20. **i** Quantitative analysis of Fig. 1h. Relative intensity of GFP signals is shown as the mean of three independent experiments ± SD, analyzed by one-way ANOVA followed by Dunnett's multiple comparison test (**$P < 0.01$). Source data are provided as a Source Data file.

perturbations induced by PR20 were compared with those induced by M9M, as the latter has previously been shown to bind to the NLS-binding site of Kapβ2[28]. As represented by peaks #63 and #69, several resonances show significant perturbations induced only by PR20, suggesting that these resonances are derived from regions dedicated for the binding of PR20 (Fig. 4). Likewise, we found several other resonances that were perturbed only by M9M (e.g., Peak #2 and #19), which can be derived from the regions dedicated to the binding of NLS (Fig. 4 and Supplementary Fig. 4a). Interestingly, several of the perturbed resonances (e.g., Peak #40 and #150) induced by the addition of PR20 were also perturbed by the binding of M9M, which suggest that these resonances were derived from regions responsible for the binding of both PR20 and NLS (Fig. 4 and Supplementary Fig. 4a). Although the perturbations could have been caused by allosteric effect, the shared perturbations indicate that PR20 partially occupy the NLS-binding site of Kapβ2. The additional reference experiment using FUS (1-500, FUS-ΔNLS) show that perturbed resonances induced by FUS-ΔNLS (e.g., Peak #37 and #173) were not perturbed by PR20, indicating that the binding site for FUS-ΔNLS does not overlap with that for PR20 (Fig. 4 and Supplementary Fig. 4b). Taken together, the NMR data suggest that the binding site of PR poly-dipeptides on Kapβ2 partially overlaps with the Kapβ2 region responsible for the recognition of NLS.

As seen in the crystal structure[11], NLS is recognized by the cavity of Kapβ2 in which several methyl-baring residues, including isoleucine residues (I457, I540, I642, I722, I773, I804), leucine residues (L419, L539, L767), valine residues (V643, V724), and methionine residue (M308), are located (Supplementary Fig. 5). Given that M9M binds to the NLS-binding site of Kapβ2, these residues can be attributed to methyl resonances that were perturbed by the binding of M9M (Fig. 4 and Supplementary Fig. 4a). Among these residues, L539, I540, I642, and V643 were found to be located at the negatively charged cavity of Kapβ2 (Fig. 4k and Supplementary Fig. 5). Considering the positive charge of PR poly-dipeptides, these four residues located at the negatively charged cavity of Kapβ2 may be important for its interaction with PR poly-dipeptides.

**PR poly-dipeptides compete with FUS-NLS for Kapβ2.** Strikingly, an MD simulation demonstrated that PR poly-dipeptides were recognized by the negatively charged cavity. The input structure for MD simulation was prepared by taking chain A of the crystallographic dimer from the crystal structure of Kapβ2 in complex with FUS (PDB: 5YVG) and mutating side chains of chain X to repeated PR sequence. L539, I540, I642, V643, I722,

L767, and I804 were shown to be located in the NLS-binding site of Kapβ2 (Fig. 5a, b and Supplementary Figs. 5 and 6), an observation that is highly consistent with data from NMR. Biophysical experiments corroborated by MD simulation indicate that PR poly-dipeptides partially bind to the NLS-binding site of Kapβ2.

To test whether PR poly-dipeptides compete with NLS of FUS for the NLS-binding site of Kapβ2, we performed a pull-down binding assay. FUS-NLS (501–526) fused to maltose-binding protein (MBP) (MBP:FUS-NLS) was subjected to GST:Kapβ2 to form a complex at a 1:1 ratio. We observed the release of MBP:FUS-NLS from Kapβ2 in the presence of PR poly-dipeptides (Fig. 5c), suggesting that PR poly-dipeptides compete with NLS of FUS for Kapβ2.

A series of interaction analyses using ITC, SEC-MALS, and NMR revealed that PR poly-dipeptides tightly bound Kapβ2 at a 1:1 ratio. The binding sites on Kapβ2 partially overlapped with sites used for the recognition of NLS, but they did not overlap with those used for the recognition of FUS-ΔNLS. Based on MD simulation, the amino acids close to the PR poly-dipeptide were in good agreement with those indicated by NMR experiments. These data imply the presence of a potential mechanism—that of PR poly-dipeptides toxicity toward Kapβ2 function in modifying phase transitions, in which PR poly-dipeptides interfere with the interaction between Kapβ2 and NLS of FUS (Fig. 6).

## Discussion

We showed that PR/GR poly-dipeptides inhibited the Kapβ2 function as a modifier of RBPs phase transitions, whereas PA/GP/GA poly-dipeptides did not, and that PR poly-dipeptides target the NLS-binding site of Kapβ2, which contains a negatively charged region. Recent studies on cellular systems have revealed that arginine-rich poly-dipeptides interact with NIRs resulting in NCT deficit[24,25] and PR poly-dipeptides have selective effect on NIRs at lower concentrations than GR poly-dipeptides[24]. We showed by a series of experiments that PR poly-dipeptides compete with NLS for binding to Kapβ2, which may provide a molecular basis for the toxicity of positively charged arginine-rich poly-dipeptides toward NIRs.

Through our experiments, we found that PR poly-dipeptides target the NLS-binding site of Kapβ2, which may lead to dysregulation of phase transitions of RBPs containing proline–tyrosine NLS. Besides FUS, many hnRNPs have proline–tyrosine NLS[29] and some of them, including hnRNPA1 and hnRNPA2, are relevant to ALS[13] and interact with TDP43[30,31]. As demonstrated in Supplementary Fig. 1, PR poly-dipeptides impede Kapβ2 function to modify the phase transition of hnRNPA2 containing

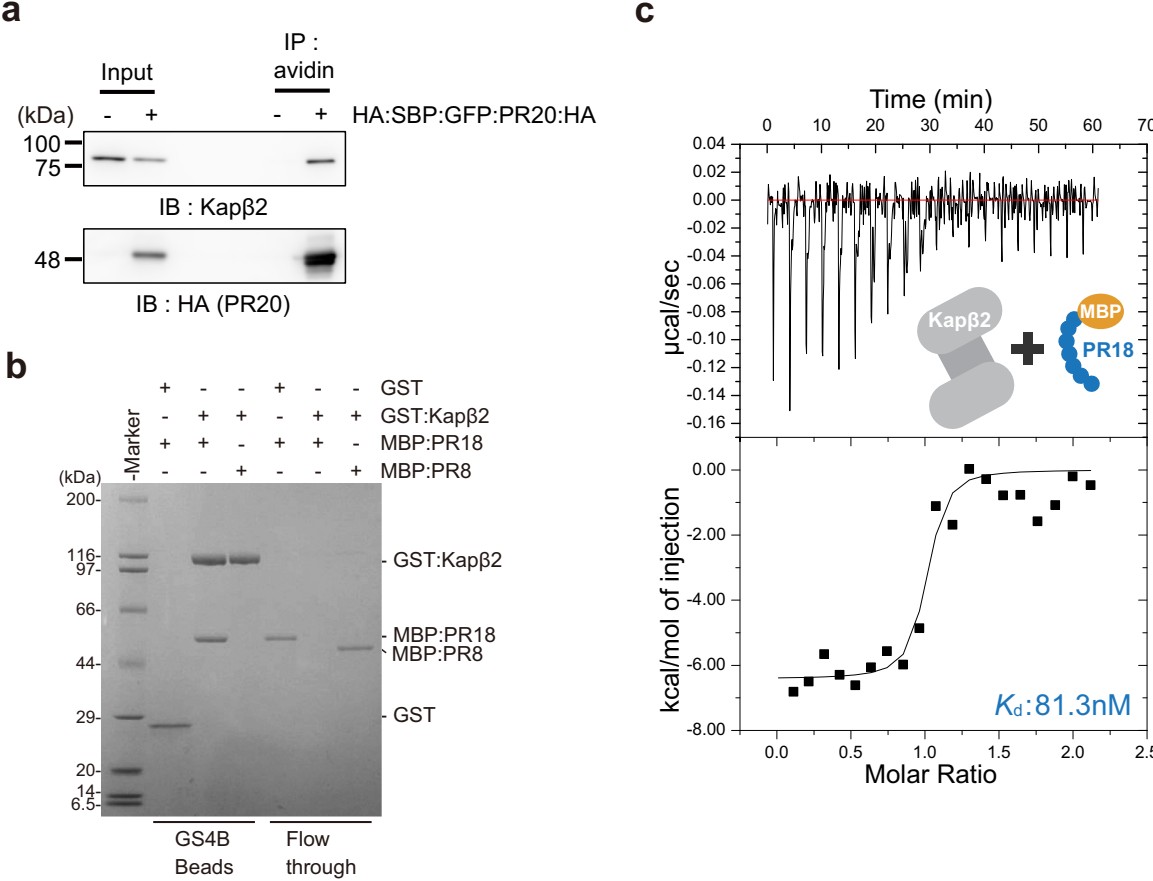

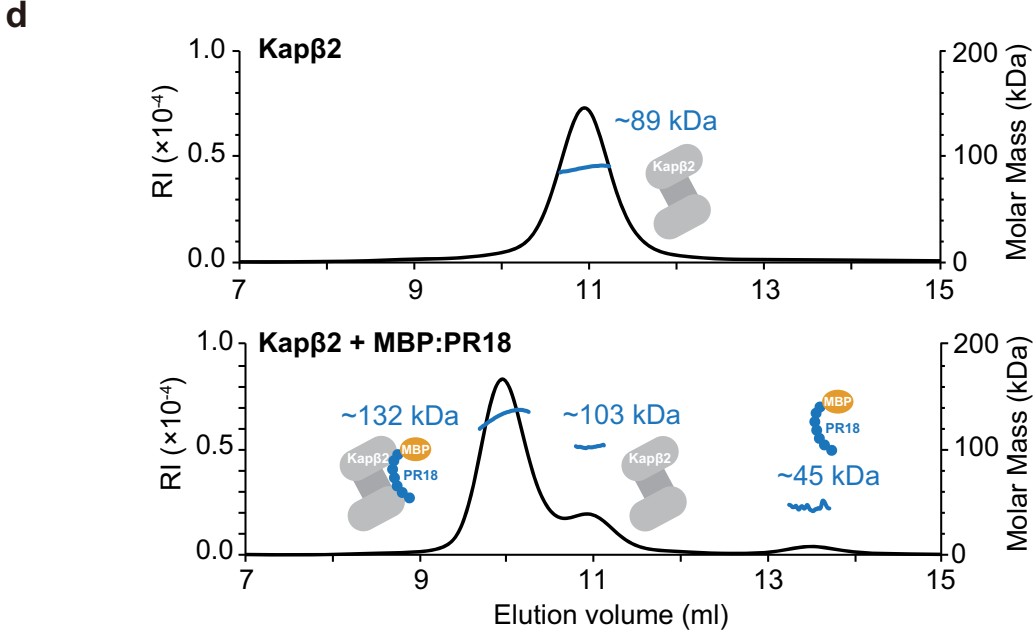

**Fig. 2 Interaction between Kapβ2 and PR poly-dipeptides. a** Immunoprecipitation showing interaction between endogenous Kapβ2 and HA:SBP:GFP:PR20:HA expressed in HeLa cells. **b** Pull-down binding assay showing interaction between GST:Kapβ2 and MBP:PR18/MBP:PR8. **c** Dissociation constant ($K_d$) measured by ITC of Kapβ2 (ΔLoop) binding to MBP:PR18. **d** SEC-MALS of Kapβ2 in the absence and presence of MBP:PR18, showing a 1:1 complex formation between Kapβ2 and MBP:PR18. The experiments (**a, b**) were independently repeated three times or more with similar results. Source data are provided as a Source Data file.

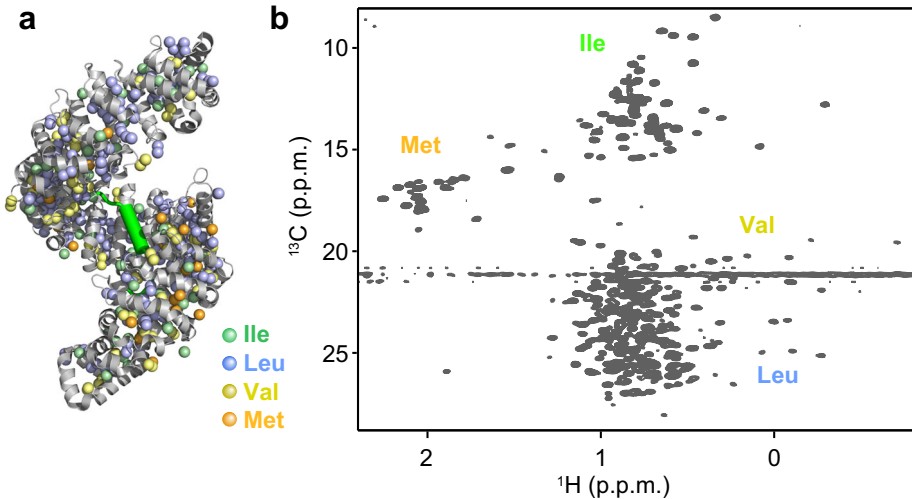

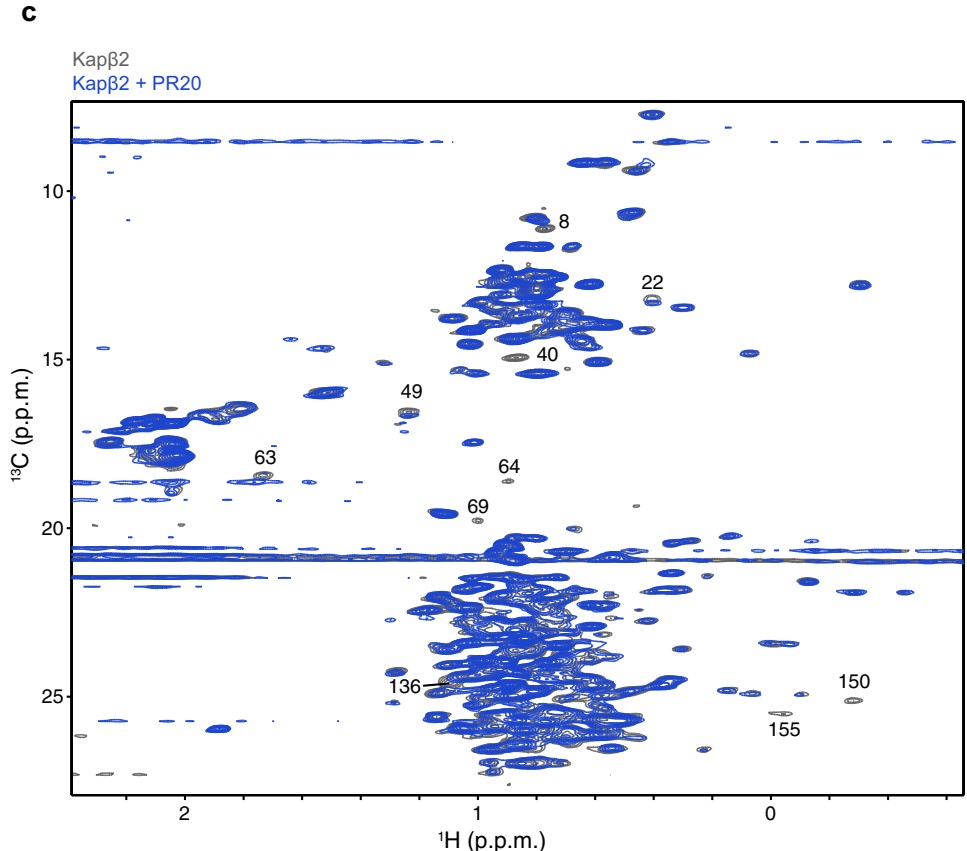

**Fig. 3 Solution NMR of Kapβ2 and interaction between Kapβ2 and PR20. a** The crystal structure of Kapβ2 in complex with NLS of FUS. NLS of FUS is presented as a green cylinder and Kapβ2 is shown as a gray ribbon, with spheres representing methyl groups of Ala (pink), Ile (green), Leu (blue), Val (yellow), and Met (orange). Kapβ2 is enriched in methyl-baring residues, indicating that sufficient information can be collected from the $^1$H-$^{13}$C-correlated methyl NMR spectra. **b** $^1$H-$^{13}$C-correlated methyl NMR spectra of [U-$^2$H; Ile-δ1-$^{13}$CH$_3$; Leu, Val-$^{13}$CH$_3$/$^{12}$CH$_3$]-labeled Kapβ2. The spectral regions for the resonances of Ile, Met, Val, and Leu methyl groups are labeled. **c** Interaction between Kapβ2 and PR poly-dipeptide investigated by solution NMR. $^1$H-$^{13}$C-correlated methyl NMR spectra of [U-$^2$H; Ile-δ1-$^{13}$CH$_3$; Leu, Val-$^{13}$CH$_3$/$^{12}$CH$_3$]-labeled Kapβ2 in the absence (gray) and presence (blue) of PR20. Significant perturbations were observed for several resonances, indicating that PR poly-dipeptides bind to specific region(s) of Kapβ2. The perturbed resonances are indicated by peak numbers. The peak numbering corresponds to those in Supplementary Fig. 3.

proline–tyrosine NLS, which might correlate with pathophysiology. Moreover, it has been shown that decreased nuclear hnRNPA3, which also has proline–tyrosine NLS, leads to an accumulation of repeat RNA and poly-dipeptides in C9-ALS/FTD[32]. If dysfunction of Kapβ2 due to PR poly-dipeptides leads to mislocalization of hnRNPA3, toxic repeat RNA and PR poly-dipeptides, products of RAN-translation, would increase, resulting in an exacerbation of C9-ALS/FTD pathophysiology.

Given existing evidence[33,34], the binding of PR poly-dipeptides to Kapβ2 might also affect the interaction between Kapβ2 and

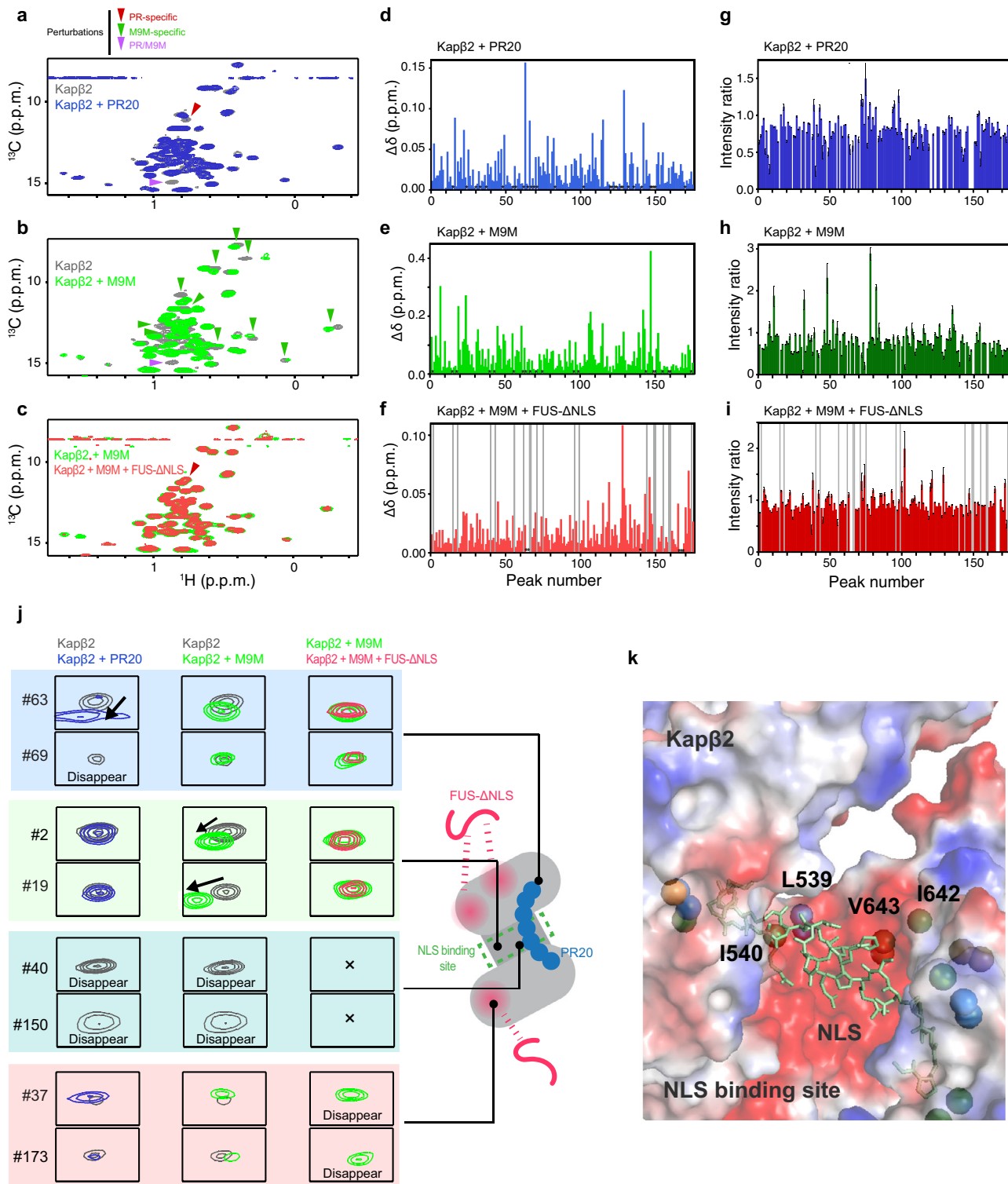

proteins with Arg/Gly/Gly (RGG) regions. A previous NMR study reported that RGG regions of the cold-inducible RBP (CIRBP-RGG) is recognized by Kapβ2[33]. The NMRs of Kapβ2 perturbed by CIRBP-RGG partially overlapped with those perturbed by PR poly-dipeptides (Supplementary Fig. 7), indicating that PR poly-dipeptides interact with the region of Kapβ2 responsible for recognizing RGG regions. The data imply that the binding of PR poly-dipeptides to Kapβ2 may also dysregulate the localization of CIRBP and thus affect the stress response system. A recent study reported that RGG regions of FUS are capable of

binding the NLS-binding site of Kapβ2[34]. The binding affinities are weaker than the interaction between PR18 and Kapβ2 (Supplementary Table 1). Thus, PR poly-dipeptides likely inhibit the interaction between RGG regions and Kapβ2 in all respects.

We showed that PR poly-dipeptides disrupt the chaperone function of NIRs for FUS/TDP43. In addition to neuronal inclusions of poly-dipeptides, aberrant cytoplasmic localization of FUS and TDP43 are observed in C9-ALS/FTD pathology[12,13]. Cytoplasmic aggregations of FUS are observed less frequently than those of TDP43 in post-mortem examinations[13]. The

**Fig. 4 Interaction between Kapβ2 and PR20/M9M/FUS-ΔNLS investigated by solution NMR.** [1]H-[13]C-correlated methyl NMR spectra of [U-[2]H; Ile-δ1-[13]CH$_3$; Leu,Val-[13]CH$_3$/C[2]H$_3$]-labeled Kapβ2 in the absence (gray) and presence (blue) of PR20 (**a**), in the absence (gray) and presence (green) of M9M (**b**), or in the presence of M9M (green) and the presence of M9M and FUS-ΔNLS (magenta) (**c**). Significant representative perturbations are indicated by arrow heads. Perturbations only seen for PR20 are indicated by red arrow heads. Perturbations only seen for M9M are indicated by green arrow heads. Perturbations common to PR20 and M9M are indicated by purple arrow heads. For clarity, only the region of the Ile methyl resonances is shown. The full-range spectra are shown in Supplementary Fig. 4 (for **a**) and Supplementary Fig. 5 (for **b** and **c**). Chemical shift differences (**d**–**f**) and intensity ratios (**g**–**i**) of the methyl resonances of Kapβ2 by the interaction with PR20 (**d**, **g**), M9M (**e**, **h**), and FUS-ΔNLS (**f**, **i**) are shown. In **d**–**f**, resonances that disappeared with the addition of a binding partner are indicated by asterisks. In **f** and **i**, resonances that disappeared with the complex formation with M9M are indicated by gray bars. In order to compare the perturbations, peaks on the spectra are labeled in peak numbers as shown in Supplementary Fig. 3. Less significant perturbations with the addition of PR poly-dipeptides than the addition of M9M can be explained by lower binding affinity of PR poly-dipeptides for Kapβ2 (Supplementary Table 1). **j** Selected views of the interaction between Kapβ2 and PR20/M9M/FUS-ΔNLS. Selected views of the representative resonance from [1]H-[13]C-correlated methyl NMR spectra of [U-[2]H; Ile-δ1-[13]CH$_3$; Leu, Val-[13]CH$_3$/C[2]H$_3$]-labeled Kapβ2 in the absence (gray) and presence of PR20 (blue), M9M (green), and M9M and FUS-ΔNLS (magenta). Graphical representation corresponds to the interaction between Kapβ2 and PR20/M9M/ FUS-ΔNLS. **k** An expanded view of the electrostatic surface potential of Kapβ2. Positive and negative surface potentials are drawn in blue and red, respectively. NLS of FUS, which is compatible with M9M, is represented as a stick model, colored pale green.

difference in tendency for aggregation between FUS and TDP43 might be derived from several factors, such as the core structures of LC-domain[35] and posttranslational modifications[9,10,35,36]. A solid-state NMR study[35] revealed that there is no hydrophobic interaction within the labile fibril core of FUS-LC and no polymorphism unlike TDP43-LC[37] or other pathogenic amyloid fibrils such as β-amyloid[38]. A recent study reported that in sporadic ALS, nuclear-cytoplasmic mislocalization of FUS was present, although cytoplasmic FUS aggregates were absent, and localization of FUS in the cytoplasm was observed prior to the appearance of TDP43 aggregates in the mouse model[39]. Hence, FUS might have a more significant role in ALS pathophysiology than previously considered. It has been reported that aggregates of TDP43 protect cells and toxic liquid phase of TDP43 harm cells[40]; therefore, further investigations are required to conclude which phase of TDP43 affects ALS pathophysiology.

Several posttranslational modifications of FUS are known to alter FUS phase transitions, including methylation, phosphorylation, and acetylation[9,10,35,36,41]. Among them, the methylation of arginine and the phosphorylation of serine are known to reduce FUS phase transitions[9,10,35,36]. RGG-rich regions of FUS are extensively methylated by protein arginine methyl transferases (PRMTs)[42]. Arginine methylation of FUS harboring disease-linked mutations decreases binding affinity to Kapβ2 and impairs nuclear transporting activity of Kapβ2[43]. Controversial results have been also reported; inclusions in ALS-FUS patients contain methylated FUS, but no methylated FUS is observed in inclusions of FTD-FUS patients, suggesting the possibility that there remain unknown pathways generating disease-related inclusions[9,43]. We revealed that PR and GR poly-dipeptides impede the chaperone activity of Kapβ2. Gly/Arg-rich sequence tends to be methylated by PRMTs[44], leading to an alternation of charge distribution. A recent study revealed that the symmetric dimethylation of GR poly-dipeptides reduces toxicity and correlates with disease duration in C9-ALS/FTD[45]. Therefore, PR poly-dipeptides might have a stronger inhibitory effect toward Kapβ2 than GR poly-dipeptides due to the evasion of methylation, although it has yet to be shown whether PR poly-dipeptides are methylated in C9-ALS/FTD.

We also demonstrated that PR poly-dipeptides tightly bind Kapβ2 in a cellular system, in addition to experiments using purified proteins. Although NIRs are abundant in a cellular environment, increased NIRs suppress the pathological protein interaction by arginine-rich poly-dipeptides[25]. A previous study revealed that overexpression of KAP104, the yeast homolog of Kapβ2, strongly suppressed the toxicity of PR poly-dipeptides[22]. These results suggest that the relation between the quantities of NIRs and those of PR poly-dipeptides might be crucial in a cellular

environment. A pathological study reported that in C9-ALS/FTD, poly-dipeptides accumulate prior to TDP43 aggregates[46]. Hence, poly-dipeptide pathology might reflect the upstream of the cascade of C9-ALS/FTD pathophysiology. Given that increased NIRs reduce the toxicity of arginine-rich poly-dipeptides and modify aberrant RBP phase transitions, our study highlights the molecular mechanism for potential C9-ALS/FTD therapeutic targets.

In summary, we showed how PR poly-dipeptides affect Kapβ2 function as a phase modifier using multiple biochemical and biophysical methods. In addition to the ability of PR poly-dipeptides to stabilize self-association of LC-domains, PR poly-dipeptides are found to bind NIRs and impede their function as phase modifiers. Our current study offers additional mechanistic insights on *C9orf72*-related neurodegeneration.

## Methods

**Constructs, protein expression, and purification.** Kapβ2 and M9M were expressed from GST fusion constructs using pGEX6P-1 and pGEX-TEV vectors, respectively. FUS and PRn proteins were expressed from MBP-fusion constructs using the pMAL-TEV vector. All recombinant proteins were expressed individually in BL21(DE3). *Escherichia coli* cells were induced with 0.5 mM isopropyl-β-ᴅ-1-thiogalactopyranoside (IPTG) for 12 h at 20 °C. Bacteria expressing Kapβ2 was lysed with a sonicator in buffer containing 50 mM Tris pH 7.5, 200 mM NaCl, 20% (v/v) glycerol, and 2 mM dithiothreitol (DTT). Kapβ2 was purified using GSH Sepharose beads (GS4B, GE Healthcare), cleaved with HRV3C protease, anion exchange chromatography (HiTrap Q HP, GE Healthcare), and gel filtration chromatography (Superdex200 16/60, GE Healthcare) in a buffer containing 20 mM HEPES pH 7.4, 150 mM NaCl, 2 mM DTT, 2 mM Mg(OAc)$_2$, and 10% glycerol. GST:M9M was purified using GSH Sepharose beads (GS4B, GE Healthcare) and gel filtration chromatography (Superdex200 16/60, GE Healthcare). To assemble the Kapβ2–M9M complex, purified Kapβ2 and GST:M9M were mixed, and GST tag was cleaved with Tobacco Etch Virus (TEV) protease. Kapβ2–M9M complex was purified by gel filtration chromatography (Superdex200 16/60, GE Healthcare) and the remaining GST was removed by GSH Sepharose beads (GS4B, GE Healthcare). MBP:FUS and MBP:PRn proteins were lysed in 50 mM Tris pH 7.5, 1.5 M NaCl, 10% glycerol, and 2 mM DTT. MBP-fusion proteins were purified by affinity chromatography, using amylose resin eluted with buffer containing 50 mM Tris pH 7.5, 150 mM NaCl, 10% glycerol, 2 mM DTT, and 20 mM Maltose. It was then further purified for MBP:PRn by cation-exchange chromatography (HiTrap SP HP, GE Healthcare) and gel filtration chromatography (Superdex200 16/60, GE Healthcare). Twenty repeats of PR/GR/PA/GP/GA with an HA epitope tag at the C terminus (PR/GR/PA/GP/GA20:HA) peptides were synthesized by SCRUM, Inc. (Japan). Primers for plasmid construction are listed in Supplementary Table 2.

**Turbidity assay and imaging of turbid solution.** Prior to adding TEV protease, we mixed 8 μM MBP:FUS, ±8 μM Kapβ2, and ±PR/GR/PA/GP/GA20:HA in a buffer containing 20 mM HEPES pH 7.4, 150 mM NaCl, 10% glycerol, 2 mM Mg(OAc)$_2$, 20 μM Zn(OAc)$_2$, and 2 mM DTT to reaction volumes of 100 μL. TEV protease was added to the premixture to a final concentration of 40 μg/mL, then incubated at 30 °C for 60 min for all the MBP-fusion protein to be digested. The solution was left to cool down to 20 °C before the measurement of OD 395 nm using a plate reader. For the imaging experiment and prior to adding TEV protease, 8 μM MBP:FUS including 0.4 μM MBP:FUS:EGFP, ±16 μM Kapβ2 or 16 μM

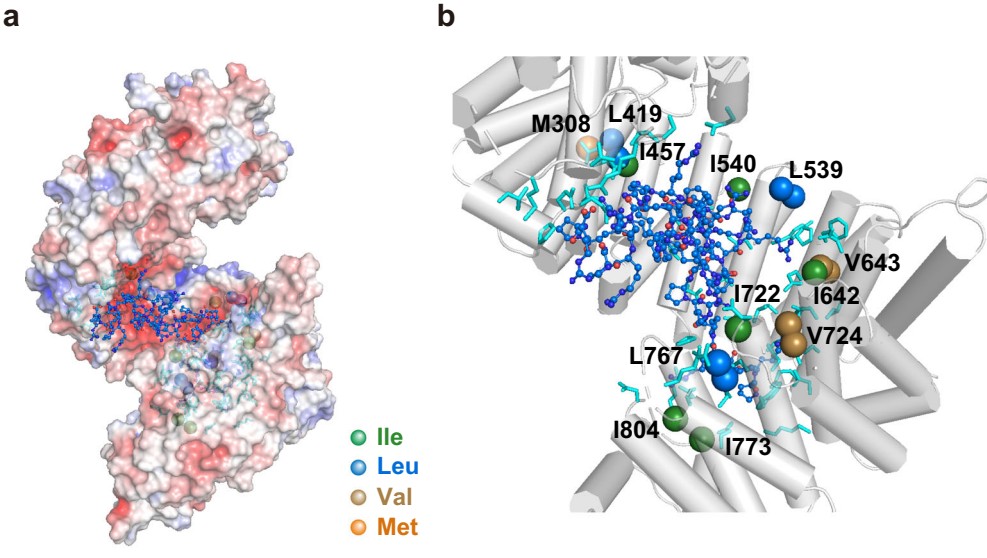

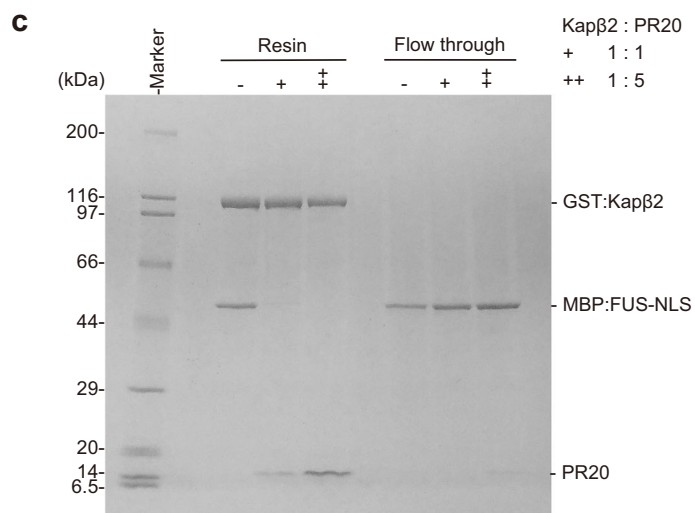

**Fig. 5 The model of interaction between PR poly-dipeptides and Kapβ2. a** Electrostatic potential of Kapβ2 and MD calculated model of PR poly-dipeptide inside of Kapβ2 cavity are shown. Blue and red colors show the positively charged and negatively charged regions, respectively. PR poly-dipeptide is colored blue. **b** Close-up view of Kapβ2 cavity. Methyl groups located close to the Kapβ2 cavity, isoleucine, leucine, valine, and methionine, are represented as spheres colored green, blue, yellow, and orange, respectively. Amino acids within 5 Å around the PR poly-dipeptide are colored cyan. **c** Pull-down binding assay showing the competition of PR20 and NLS of FUS to GST:Kapβ2. The experiment was independently repeated twice with similar results. Source data are provided as a Source Data file.

Kapβ2–M9M complex, and ±50 μM PR20:HA were mixed in a buffer containing 20 mM HEPES pH 7.4, 150 mM NaCl, 10% glycerol, 2 mM Mg(OAc)$_2$, 20 μM Zn(OAc)$_2$, and 2 mM DTT to reaction volumes of 20 μL. TEV protease was added to the premixture to a final concentration of 40 μg/mL, then incubated at room temperature for 3 h. MBP:FUS:EGFP in the solutions was imaged using a fluorescent microscope (Nikon Eclipse Ti2) equipped with a ×40, 0.6 na objective lens (Nikon) and a CMOS camera (ORCA-spark, Hamamatsu). The system was operated using the NIS-Elements software (Nikon).

**Hydrogel binding assay.** Expression plasmids for recombinant proteins (GFP/mCherry fusion LC-domain of FUS (residue 2–214, GFP/mCh:FUS-LC) and GFP/mCherry fusion LC-domain of hnRNPA2 (residue 181–341, GFP/mCh:hnRNPA2-LC)) were obtained from the Steven L. McKnight Laboratory. Expression plasmid for GFP:FUS-LC fusion NLS (residue 501–526, GFP:FUS-LC:NLS), mCh fusion LC-domain of TDP43 (residue 262–414, mCh:TDP43-LC), and GFP-tagged cNLS (residue 78–99) fusion TDP43-LC (GFP:cNLS:TDP43-LC) were constructed using In-Fusion HD Cloning Kit (Takara Bio, Inc.). GFP/mCh:FUS-LC, GFP:FUS-

LC:NLS, and GFP/mCh:hnRNPA2-LC were expressed in *E. coli* BL21(DE3) cells with 0.5 mM IPTG at 20 °C overnight and purified by Ni-NTA Agarose (FUJIFILM Wako Pure Chemical Corporation), as described in a previous study[47]. GFP:cNLS:TDP43-LC were expressed with 1.0 mM IPTG at 37 °C for 3 h and mCh:TDP43-LC were expressed with 0.5 mM IPTG at 16 °C overnight. GFP:cNLS:TDP43-LC and mCh:TDP43-LC were purified by Ni-NTA as described in a previous study[48] with some modifications. For the purification of mCh:TDP43-LC/GFP:cNLS:TDP43-LC, cells were lysed in buffer containing 25 mM Tris-HCl pH 7.5, 200 mM NaCl, 2 M/4 M urea, 10 mM β-mercaptoethanol, and Protease Inhibitor Cocktail Tablets (Sigma), washed with a buffer (25 mM Tris-HCl pH 7.5, 200 mM NaCl, 2 M Urea, 10 mM β-mercaptoethanol, and 20 mM imidazole), and eluted from the resin with an elution buffer (25 mM Tris-HCl pH 7.5, 200 mM NaCl, 2 M Urea, 10 mM β-mercaptoethanol, and 300 mM imidazole). Hydrogel droplets of mCh:FUS-LC, mCh:hnRNPA2-LC, and mCh:TDP43-LC were prepared as reported in a previous study[47]. For hydrogel binding assays, purified GFP-fused proteins were diluted to 1 μM in the buffer (20 mM Tris-HCl pH 7.5, 150 mM NaCl, 20 mM β-mercaptoethanol, 0.1 mM phenylmethylsulfonyl fluoride, and 0.5 mM EDTA) and pipetted onto a hydrogel dish. After overnight

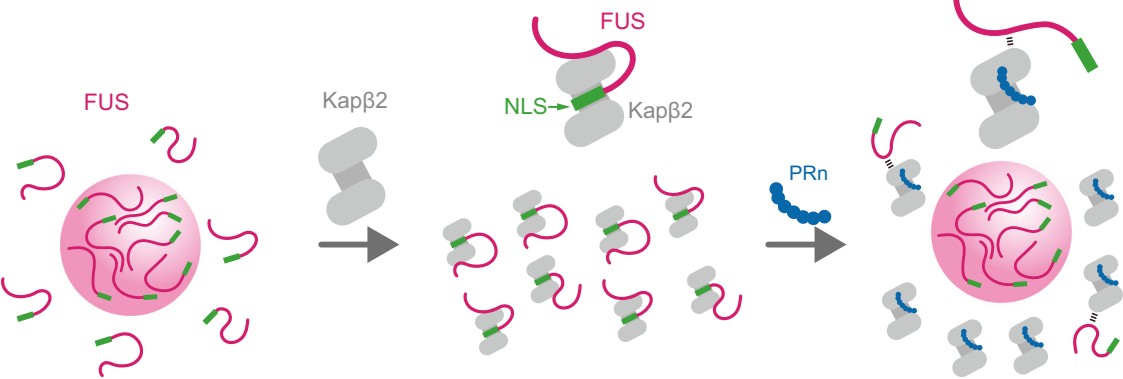

**Fig. 6 Graphical abstract for the model of interaction between PR poly-dipeptides and Kapβ2.** FUS is prone to self-associate (left). Kapβ2 modifies a phase transition of FUS by recognizing NLS (middle). PR poly-dipeptides partially bind to the NLS-binding site of Kapβ2 and impede its ability (right).

incubation, horizontal sections of the droplets were scanned with excitation wavelengths on a confocal microscope (FLUOVIEW FV3000, OLYMPUS). Relative intensity of GFP signals across the hydrogel droplets were measured in triplicate by using the profile plot mode in ImageJ. The values are shown as mean ± SD. Statistical analyses were performed using Graphpad Prism version 7.

**Immunoprecipitation**. The pcDNA5/FRT/TO-HA:SBP plasmid was generated by inserting the PCR-amplified HA:SBP sequence into the pcDNA5/FRT/TO vector (Thermo Fisher Scientific) between the HindIII and BamHI sites. The HA:SBP:GFP:PR20:HA plasmid was constructed by ligating the GFP:PR20:HA sequence into the pcDNA5/FRT/TO-HA-SBP between the EcoRV and XhoI sites. The HA:SBP:Kapβ2 plasmid was constructed by ligating the Kapβ2 sequence into the pcDNA5/FRT/TO-HA:SBP between the BamHI and NotI sites. HeLa cells were cultured in Dulbecco's modified Eagle medium/10% fetal bovine serum at 37 °C with 5% $CO_2$. Cells were transfected with plasmids using Lipofectamine 3000 (Thermo Fisher Scientific) according to the manufacturer's instructions. HeLa cytosol extracts were prepared by Cytoplasmic Extraction buffer (CE; 10 mM HEPES pH 7.9, 60 mM KCl, 1 mM EDTA, 0.075% NP40, 1 mM DTT, and cOmplete™ EDTA-free protease inhibitor [Sigma-Aldrich]) for 10 min on ice and then separated from nuclear by centrifugation at $500 \times g$ for 10 min. The resultant extracts were cleared by centrifugation at $20,400 \times g$ for 10 min. The supernatant containing the HeLa cytosol extracts were mixed with the Streptavidin Mag Sepharose (GE Healthcare) and rotated at 4 °C overnight. The beads were finally washed five times with CE buffer. The anti-Kapβ2 (sc-166127, 1 : 1000) antibody was obtained from Santa Cruz Biotechnology. The anti-HA (561, 1 : 10,000) antibodies were obtained from MBL. The anti-Mouse IgG H&L (Horseradish peroxidase, HRP) (ab6789, 1 : 10,000) and anti-Rabbit IgG H&L (HRP) (ab6721, 1 : 10,000) antibodies were purchased from Abcam. HeLa cells were purchased from ATCC.

**Pull-down binding assay**. In-vitro pull-down binding assays showing the interaction between PR poly-dipeptides and Kapβ2 were performed using GST:Kapβ2, GST:PR18, GST:Importinα, or GST:Importinβ1 immobilized on GSH sepharose beads (GE Healthcare). Four micrograms of GST proteins were immobilized on 30 μL of beads. GST protein beads were incubated with equal molar protein, either MBP:PR18, MBP:PR8, Kapβ2, BSA, MBP, EGFP, or Lysozyme for 20–30 min and washed three times with a buffer containing 20 mM HEPES pH 7.4, 150 mM NaCl, 10% glycerol, 2 mM Mg(OAc)₂, and 2 mM DTT. For the competitive binding assay, 16 μg GST:Kapβ2 were immobilized on 160 μL beads and washed three times with the buffer before the beads were aliquoted into four tubes and incubated with equal molar MBP:FUS-NLS (residue 501–526) ± PR20:HA for 20–30 min. Unbound proteins were washed three times with the buffer. Bound proteins and unbound proteins were separated by SDS-polyacrylamide gel electrophoresis and stained with Coomassie Brilliant Blue.

**Isothermal titration calorimetry**. ITC experiments were performed with a Malvern iTC200 calorimeter (Malvern Instruments). Proteins were dialyzed overnight against buffer containing 20 mM HEPES pH 7.4, 150 mM NaCl, 10% glycerol, and 2 mM β-mercaptoethanol. MBP:PR18 (200 μM) were titrated into the sample cell containing 20 μM Kapβ2 (ΔLoop, residues 321–371 were replaced with GGSGGSGS linker). ITC experiments were performed at 25 °C with 19 rounds of 2 μL injections. Data were analyzed using Origin software.

**Size-exclusion chromatography with multi-angle light scattering**. SEC-MALS was measured using DAWN HELEOS8+ (Wyatt Technology Corporation) downstream of a Shimadzu liquid chromatography system connected to Superdex200 10/300 GL (GE Healthcare) gel filtration column. The differential refractive

index (Shimadzu Corporation) downstream of MALS was used to obtain protein concentration. The column was equilibrated with a running buffer containing 20 mM HEPES pH 7.4, 150 mM NaCl, 2 mM MgCl₂, 10% glycerol, and 2 mM β-mercaptoethanol. Flow rate was set to 0.5 mL min⁻¹ and 100 μL of the sample was injected. Kapβ2 at a concentration of 28 μM was injected in the absence and presence of 55 μM of MBP:PR18. The data were analyzed with ASTRA version 7.0.1 (Wyatt Technology Corporation).

**Analytical ultracentrifugation (AUC)**. AUC measurements were conducted with ProteomeLab XL-I (Beckman Coulter). Three solutions, 28 μM Kapβ2, 28 μM MBP:PR18, and the mixture of 28 μM Kapβ2 + 28 μM MBP:PR18, were prepared in the buffer containing 20 mM HEPES pH 7.4, 150 mM NaCl, 2 mM MgCl₂, 2 mM DTT, and 10% glycerol. The solutions were filled in 1.5 mm path-length titanium cells (Nanolytics). The measurements were performed with Rayleigh-interference optics at rotor speed of $125,000 \times g$ (at 70 mm radius). The temperature was set at 25 °C. The weight concentration distribution as a function of sedimentation coefficient $c(s_{20,w})$ and frictional ratio $f/f_0$ were computed with SEDFIT[49] and Igor Pro software. The sedimentation coefficient was normalized to be the value at 20 °C in pure water $s_{20,w}$. The molecular weight $M$ was calculated with the corresponding peak value of $s_{20,w}$ and $f/f_0$[49].

**Expression and purification of isotopically labeled Kapβ2**. The Kapβ2 expression plasmid was transformed into BL21(DE3) cells. The protein samples with ¹H,¹³C-labeled methyl groups in deuterium background were prepared as described in a previous study[26]. The cells were grown in medium with ¹⁵NH₄Cl (2 g L⁻¹, CIL) and ²H₇-glucose (2 g L⁻¹, CIL) in 99.9% ²H₂O (Isotec). The precursors for methyl groups of Ile, Val, and Leu, α-ketobutyric acid (50 mg L⁻¹) and α-ketoisovaleric acid (80 mg L⁻¹), and [¹³CH₃] methionine (50 mg L⁻¹) were added to the culture 1 h before the addition of IPTG. Protein expression was induced by the addition of 0.5 mM IPTG at $OD_{600} \sim 0.6$, followed by ~16 h of incubation at 25 °C. Cells were collected and re-suspended in the lysis buffer containing 50 mM HEPES pH 7.4, 150 mM NaCl, 20% glycerol, 2 mM DTT, and 2 mM EDTA. Cells were disrupted by a sonicator and centrifuged at $38,900 \times g$ for 45 min. Proteins were purified using GS4B resin (GE Healthcare) and eluted with the buffer containing 20 mM HEPES pH 7.4, 20 mM NaCl, 2 mM EDTA, 10% glycerol, and 30 mM GSH. The GST tag was removed by HRV3C protease at 4 °C for ~16 h. The protein was further purified by anion exchange using HiTrap Q FF (GE Healthcare) with the buffer containing 20 mM Imidazole pH 6.5, 20–1000 mM NaCl, 2 mM EDTA, 2 mM DTT, and 20% glycerol, followed by gel filtration using Superdex200 16/60 (GE Healthcare) with the buffer containing 20 mM HEPES pH 7.4, 150 mM NaCl, 2 mM MgCl₂, 2 mM DTT, and 2% glycerol. Protein concentration was determined spectrophotometrically at 280 nm using a corresponding extinction coefficient.

**NMR experiments**. The isotopically labeled Kapβ2 was prepared in the NMR buffer containing 20 mM ²H-Tris pH 7.4, 150 mM NaCl, 2 mM MgCl₂, 2 mM DTT, and 2% glycerol, concentrated to 20 μM. NMR experiments were performed on Bruker 600 MHz and 800 MHz NMR operated with Topspin software at 20 °C. Perturbation of the side chain methyl resonances were each monitored using ¹H-¹³C heteronuclear multiple quantum coherence. Spectra were processed using the NMRPipe software[50]. Data analyses were performed on Olivia software (https://github.com/yokochi47/Olivia). The perturbations of the resonances were evaluated based on chemical shift change or intensity change. Chemical shift perturbations for the methyl groups were calculated based on the following equation:

$$\triangle \delta = \sqrt{\left(\frac{\triangle \delta_H}{\alpha}\right)^2 + \left(\frac{\triangle \delta_C}{\beta}\right)^2} \qquad (1)$$

where $\Delta\delta_H$ and $\Delta\delta_C$ are the respective chemical shift changes of $^1$H and $^{13}$C by the addition of the ligand, and $\alpha$ and $\beta$ are the respective chemical shift distributions of $^1$H and $^{13}$C of methyl groups as reported in the Biological Resonance Data Bank (http://www.bmrb.wisc.edu)[51]. Errors for the intensity ratio were estimated based on the peak intensity and noise level.

In order to investigate the interaction between Kapβ2 and PR poly-dipeptide by NMR, the NMR spectra of the isotopically labeled Kapβ2 in the absence and presence of the synthetic peptide of PR20:HA were measured. To investigate the interaction between Kapβ2 and M9M, the Kapβ2–M9M complex was prepared in the following manner. Purified GST:M9M protein was added to the isotopically labeled Kapβ2, followed by the removal of the GST tag by TEV protease at room temperature for 2 h and then at 4 °C for ~16 h before it was buffer-exchanged into the NMR buffer using Amicon Ultra-4 50k (Merck). To investigate the interaction between Kapβ2 and FUS-ΔNLS by NMR, the NMR spectra of isotopically labeled Kapβ2 in complex with M9M were measured in the absence and presence of FUS-ΔNLS.

**MD simulation**. MD simulations were performed by Maestro version 11.0.014 (Schrödinger) with Desmond program (D. E. Shaw Research)[52]. Protonation states of titratable residues at pH 7.0 were determined by the PROPKA program[53,54]. The protein and PR poly-dipeptide were solvated with a rectangular box of TIP3P[55] water molecules and neutralized by adding three sodium ions. The simulation system was equilibrated with the default protocol of the Desmond program. The protein backbone (except for the PR poly-dipeptide) was restrained by a force constant of 100 kcal (mol Å)$^{-1}$ to keep the original Kapβ2 structure during simulations. After a 5 ns relaxation simulation, a 100 ns MD simulation was performed under the conditions where equilibration of simulation was performed using the isothermal–isobaric ensemble (NPT). The force field used was OPLS-2005[56], the van der Waals interaction between atoms separated by over 10 Å were cut off, and long-range interactions were computed with the particle mesh Ewald method. The thermostat and barostat used were Nosé–Hoover chain[57] at 300 K and Martyna–Tobias–Klein[58] at 1 atm with relaxation of 1 and 2 ps. The RESPA integrator[59] was used with Fourier-space electrostatics computed every 6 fs and all remaining interactions computed every 2 fs. Surface electrostatic potential was calculated using Adaptive Poisson–Boltzmann Solver (version 2.1). The structure was drawn with PyMol program (version 2.3.0.). The electrostatic potential of Kapβ2 was calculated by PyMOL APBS tool at 0.15 M of monovalent salt (±1) and represented within a ± 5 kT/e range.

**Reporting summary**. Further information on research design is available in the Nature Research Reporting Summary linked to this article.

## Data availability
The data supporting the findings of this study are available from the corresponding authors upon reasonable request. PDB ID referred to in this study is as follows: 5YVG and 4FDD. Source data are provided with this paper.

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

## Acknowledgements

This work was supported by grants from AMED [JP20dm0307032 and JP21wm0425004 to E.M.; JP21ek0109437 to T. Saio; and JP21ek0109558 to T.Y.], JSPS KAKENHI [JP17H07031 and JP20H03199 to E.M.; JP18H06202, JP19K21306, and JP20K16583 to H.N.; JP19H04945, JP19K06504, JP18H05229, JP17H05657, JP17H05867, JP20KK0156, and JP20H03199 to T. Saio; JP19H05769 to K.I.; JP19K16060 to T.Y.; JP20K06493 to T.M.; JP19K16088 and JP21K15051 to K.K.; JP19KK0071 and JP20K06579 to R.I.; JP18H05229, JP18H05534, and JP18H03681 to M. Sugiyama; JP20K06527 to T.O.; JP20H00327 to N.K.; JP21K06031 to S.T.F.; JP18H02391 to M. Sato; JP20H05925 and JP21H04763 to H.T.; JP18K06094 to H.M.; JP19K23976 to M.N.; JP19K17043 to T. Shiota; JP21K15032 to S.K.; JP19K17044 to N.I.; and JP19K07978 to T.K.], JST FOREST Program JPMJFR204W to T. Saio; JST CREST Program JPMJCR1762 to N.K.; the Cooperative Research Project Program of Life Science Center for Survival Dynamics, Tsukuba Advanced Research Alliance (TARA Center), University of Tsukuba, Japan (Grant number: 181107, 201904, 202007) to S.T.F.; Joint Research Programs, the Institute of Advanced Medical Sciences, Tokushima University to E.M., T. Saio, and K.I.; Takeda Science Foundation to E.M. and T. Saio; Kanzawa Medical Research Foundation to E.M.; Uehara Memorial Foundation to E.M., T.Y., and S.K.; Nakatomi Foundation to E.M.; Konica Minolta Science and Technology Foundation to E.M.; Naito Foundation to E.M.; MSD Life Science Foundation to E.M.; Mochida Memorial Foundation for Medical and Pharmaceutical Research to E.M.; SENSHIN Medical Research Foundation to E.M.; Terumo Foundation for Life Sciences and Arts to E.M.; Nara Kidney Disease Research Foundation to E.M.; Novartis Research Grants to E.M., H.N., and K.S.; Nara Medical University Grant-in-Aid for Collaborative Research Projects to K.S.; Akiyama Life Science Foundation to T. Saio; Northern Advancement Center for Science and Technology to T. Saio; The Sumitomo Foundation to T. Saio; Astellas Foundation for Research on Metabolic Disorders to T. Saio; Senri Life Science Foundation to T. Saio; The Nakajima Foundation to T. Saio; The Asahi Glass Foundation to T. Saio; Daiichi Sankyo Foundation of Life Science to T.Y.; The Nakabayashi Trust For ALS Research Grants-in-Aid to T.Y.; Integrated Research Consortium on Chemical Sciences to Y.A.; Izumi Science and Technology Foundation to Y.A.; Tokyo Biochemical Research Foundation to S.K.; and by unrestricted funds provided to E.M. from Dr. Taichi Noda (KTX Corp., Aichi, Japan) and Dr. Yasuhiro Horii (Koseikai, Nara, Japan). This work was also partially supported by the project for Construction of the basis for the advanced materials science and analytical study by the innovative use of quantum beam and nuclear sciences in KURNS. We thank Steven L. McKnight, Yuh-Min Chook, Charalampos Kalodimos, Masato Kato, Benjamin Tu, Kentaro Shiraki, and Keren-Happuch E. for their critical reading of the manuscript. We thank Fumio Takahashi for his help with microscopic imaging and Yoichi Takeda for his help with ITC data collection. We thank Hiroyuki Kumeta for his help in setting up NMR experiments. The NMR experiments were performed at Hokkaido University Advanced NMR Facility, a member of NMR Platform.

## Author contributions

H.N., H.K., A.F., T. Saio, T.Y., and E.M. designed the research. H.N., H.K., A.F., T.U., Y.A., M.N., M.H., S.K., M.M., Y.S., T.N., T.M., N.M., K.M., R.I., T. Saio, and T.Y. performed research. H.N., H.K., A.F., S.K., Y.A., T. Saio, and T.Y. analyzed the data. H.N., Y.A., T. Saio, T.Y., and E.M. wrote the paper. T. Shiota, P.W., R.N., I.N., T.K., M. Sugiyama, T.O., N.K., S.T.F., M. Sato, H.T., S.N, O.S., K.I., H.M., and K.S. helped to analyze and interpret the data, and critically revise the manuscript. T. Saio, T.Y., and E.M. conceptualized the study, developed the study design, supervised the authors throughout the study, and provided expertise in manuscript preparation. All authors read and approved the final manuscript.

## Competing interests

The authors declare no competing interests.
