## [Peer Review File · Nature Communications]

REVIEWER COMMENTS

Reviewer #1 (Remarks to the Author):

The submitted manuscript by Nanaura et. al makes the interesting observation that polyPR repeat peptides derived from C9orf72 attenuates the ability of nuclear import protein KapB2 to chaperone phase separation of PY NLS containing low-complexity proteins. The results presented here add an interesting wrinkle to the trio of papers published in 2018 all similarly reporting the impact of KapB2. A major strength of the manuscript is the rigorous characterization of polyPR directly interacting with KapB2 and displacing (or outcompeting) the PY NLS. The combination if ITC, SEC-MALS, pull-down experiments and NMR spectroscopy are very convincing. We can no add "binding to importin" to the catalog of ways that C9orf72 dipeptides can potentially impact cellular homeostasis.

Concerns: The interrogation of phase separation feels incomplete. It is clear from the authors model that polyPR binds to KapB2 and out competes the PY NLS. The mechanism is thus fairly straight forward that, polyPR will sequester KapB2 and prevent its interaction with FUS or hnRNPA2 droplets. I believe the authors are trying to demonstrate that polyPR behaves analogously to the M9 NLS peptide. However, I would like to see a more complete mapping of the phase space between FUS/A2, KapB2 and polyPR. For example, at what concentration ratios does polyPR inhibit chaperoning of FUS phase separation? How well does this compare to the relative affinities of the PY NLS and polyPR to KapB2? I think the addition of this data would greatly strengthen the message.

The discussion is far too brief. The interaction of KapB2 and polyPR adds to the narrative of C9orf72 peptides disrupting nuclear transport. I would like to see a complete discussion of how the results presented here fit into the greater canon of C9orf72 dipeptide pathology. Is inhibiting the chaperoning effect on PY containing low complexity proteins the only impact or could this further perturb nuclear transport? Could this mechanism influence other proposed C9orf72 pathologies?

The same can be said for the introduction. This is a rich field and the problem deserves more setup.

Other comments:

I would suggest that the manuscript might read better if the authors lead with the data indicating an interaction between KapB2 and polyPR and conclude with the impact on FUS/A2. As mentioned above, given the interaction this effect could be expected at the concentrations used. As far as I'm aware, there is no proposed direct interaction between polyPR and KapB2 or any proposed direct effect of polyPR on accumulation of LCDs in cytoplasmic condensates.

I would suggest reorganizing the main figures and supplement. I would like to see the NMR data showing complete chemical shift perturbations in the main text. In my opinion the pull down data follows from the ITC and SEC MALS and need not be in the main text.

I would like to see a more complete mapping of the interaction surface of polyPR with KapB2 in the main text. The authors have all of the data, it simply needs to be graphically shown in the main text. This would greatly improve the message.

In summary, I believe the direct interaction between C9orf72 dipeptides and KapB2 is an important finding and the authors rigorously characterize this. I believe these results should be published. Hopefully my suggestions help the authors improve the final result.

Reviewer #2 (Remarks to the Author):

Pathologic DPR production is a molecular hallmark of C9orf72 ALS/FTD. While DPRs have previously

been shown to interfere with nucleocytoplasmic transport, it is unknown precisely how they may interfere with karyopherin mediated transport and/or chaperoning of RNA binding proteins. Nanaura et al use a series of in vitro assays to evaluate whether the poly(PR) DPR protein disrupts the association between KapB2 and the RNA binding protein FUS. While the manuscript presents interesting data from a protein interaction perspective, the findings are not particularly novel. Moreover, the implications for cellular pathophysiology in disease are completely unclear and would greatly strengthen the novelty and overall impact of this manuscript. In its current form, I can not recommend this paper for publication at Nature Communications. However, specific comments that if addressed would make this manuscript suitable for consideration at Nature Communications can be found below.

1. I find it rather odd that the authors focused on only PR. However, multiple DPR species have been shown to disrupt NCT (including a recent bioRxiv paper regarding GA sequestration of karyopherins). Thus, the reason for selecting just PR remains unclear and additional DPR species would make for a nice comparison.
2. It is also unclear why the authors focus on FUS. Is FUS pathology a prominent feature of C9 ALS/FTD? The in vitro binding experiments suggest that PR can associate with multiple karyopherins, including those that may be implicated in the transport and/or chaperoning of TDP-43 which is a prominent pathological feature of C9 ALS/FTD. Thus, the authors should additionally investigate this to increase relevance to disease pathophysiology.
3. My largest and most important concern is that this entire manuscript is based on in vitro data. Given the conflicting evidence in the literature regarding the effects of DPRs on NCT, it is essential that the authors demonstrate relevance of their findings to a living cellular system. While in a test tube PR may disrupt phase transitioning of FUS via KapB2 interference, does this phenomenon actually occur in a real cellular environment in the context of multiple karyopherins and NCT factors, which are often present in excess quantities.

Minor Comments

1. Manuscript is hard to follow in places due to the fact that some sentences appear to be incomplete.
2. Stats are missing from multiple graphs.

Reviewer #3 (Remarks to the Author):

In this study, together with experiments, the authors used molecular dynamics (MD) simulations to further prove the effective binding of proline:arginine poly-dipeptides in the predicted binding site.

I will only comment on the MD part. In this study, MD simulations are used in order to sustain the finding that poly-dipeptides can directly bind to the predicted binding site. Certainly, MD is a well-suited technique to study problems like the one tackled in this study. However, these simulations last 15ns, which is far too short nowadays. Any potential unbinding event would not be observable in such a short run. These simulations do not meet the current standards of MD simulations (typically hundreds of ns, at least, for a problem and model system like the one contained in this study). Indeed, a reasonable solution would be to extend the simulations to reach at least about 100ns). Moreover, the authors should show at least the root mean square deviation (RMSD) of both the protein and the poly-dipeptide, distinctly, during the simulations. This would reinforce the data and somehow sustain a bit more their findings.

The authors should specify:

- the force fields used to model the protein, the poly-dipeptides as well as the sodium ions;
- How the electrostatic potential was calculated on the protein surface.

In summary, there is very little here that can be said in favor of such MD simulations. Details on the

model system and parameteres are missing. The analysis of (only) 15ns is poor, and my sense is that MD here is very marginal in the overall story.

Point-by point response to reviewers' comments

Reviewer #1:

The submitted manuscript by Nanaura et. al makes the interesting observation that polyPR repeat peptides derived from C9orf72 attenuates the ability of nuclear import protein KapB2 to chaperone phase separation of PY NLS containing low-complexity proteins. The results presented here add an interesting wrinkle to the trio of papers published in 2018 all similarly reporting the impact of KapB2. A major strength of the manuscript is the rigorous characterization of polyPR directly interacting with KapB2 and displacing (or outcompeting) the PY NLS. The combination of ITC, SEC-MALS, pull-down experiments and NMR spectroscopy are very convincing. We can now add "binding to importin" to the catalog of ways that C9orf72 dipeptides can potentially impact cellular homeostasis.

Concerns: The interrogation of phase separation feels incomplete. It is clear from the authors model that polyPR binds to KapB2 and out competes the PY NLS. The mechanism is thus fairly straight forward that, polyPR will sequester KapB2 and prevent its interaction with FUS or hnRNPA2 droplets. I believe the authors are trying to demonstrate that polyPR behaves analogously to the M9 NLS peptide. However, I would like to see a more complete mapping of the phase space between FUS/A2, KapB2 and polyPR. For example, at what concentration ratios does polyPR inhibit chaperoning of FUS phase separation? How well does this compare to the relative affinities of the PY NLS and polyPR to KapB2? I think the addition of this data would greatly strengthen the message.

We appreciate your suggestions and have performed additional hydrogel binding assay (Fig.1h, i). The results showed that PR poly-dipeptides inhibited the chaperone function of Kap β 2 dose-dependently. We revealed by ITC that PR poly-dipeptide bound to Kap β 2 at K_d value of 81.3nM (Fig. 2c) and compared the affinity with that of FUS toward Kap β 2 (page 4, line 2-4). We have also updated the table to show the relative affinity of Kap β 2 towards segments of FUS with various length (Supplementary Table 1).

The discussion is far too brief. The interaction of KapB2 and polyPR adds to the narrative of C9orf72 peptides disrupting nuclear transport. I would like to see a complete discussion of how the results presented here fit into the greater canon of C9orf72 dipeptide pathology. Is inhibiting the chaperoning effect on PY containing low complexity proteins the only impact or could this further perturb nuclear transport? Could this mechanism influence other proposed C9orf72 pathologies?

Thank you for your comments. As suggested, we have reorganized our discussion section. Our current discussion considers how the inhibitory effects of PR poly-dipeptides towards Kap β 2 might lead to aberrant phase separation of various RNA-binding proteins (RBPs) containing proline-tyrosine NLS, an exacerbation of repeat RNA toxicity through hnRNPA3, and an inhibition of the interaction between Kap β 2 and RGG-rich domain of RBPs (page 5, line 31-43, page 6, line 1-9).

The same can be said for the introduction. This is a rich field and the problem deserves more setup.

As recommended, we have rewritten the introduction section and provided a description of the problems of C9-ALS/FTD pathophysiology with additional references (page 2, line 15-45, page 3, line 1-2).

Other comments:

I would suggest that the manuscript might read better if the authors lead with the data indicating an interaction between KapB2 and polyPR and conclude with the impact on FUS/A2. As mentioned above, given the interaction this effect could be expected at the concentrations used. As far as I'm aware, there is no proposed direct interaction between polyPR and KapB2 or any proposed direct effect of polyPR on accumulation of LCDs in cytoplasmic condensates.

We have revised our manuscript as suggested by the reviewer. In the current manuscript, we have specified the concentrations used in the experiments (Fig. 1f-h). Additionally, we performed immunoprecipitation, showing the direct interaction between PR poly-dipeptides and Kap β 2 in cellular environment (page 3, line 33-36, Fig.2a).

I would suggest reorganizing the main figures and supplement. I would like to see the NMR data showing complete chemical shift perturbations in the main text. In my opinion the pull down data follows from the ITC and SEC MALS and need not be in the main text.

Thank you for your recommendation. We have reorganized our main figures and supplementary figures; complete chemical shift perturbations are now shown in Fig. 4. We have added results from immunoprecipitation assay showing interaction between PR poly-dipeptides and Kap β 2 in cellular environment (Fig.2a), and we have opted to keep pull-down assay data with purified protein in the main figure (Fig. 2b) followed by ITC and SEC-MALS.

I would like to see a more complete mapping of the interaction surface of polyPR with KapB2 in the main text. The authors have all of the data, it simply needs to be graphically shown in the main text. This would greatly improve the message.

As suggested, we have included a description of the NMR data with peak numbers showing the interaction between PR poly-dipeptides and Kap β 2 within the main text (page 4, line 14-36).

In summary, I believe the direct interaction between C9orf72 dipeptides and KapB2 is an important finding and the authors rigorously characterize this. I believe these results should be published. Hopefully my suggestions help the authors improve the final result.

We thank Reviewer 1 again for taking time to provide constructive comments on our manuscript. We have tried our best to address all of the comments and believe that the revisions have significantly improved our manuscript.

Reviewer #2:

Pathologic DPR production is a molecular hallmark of C9orf72 ALS/FTD. While DPRs have previously been shown to interfere with nucleocytoplasmic transport, it is unknown precisely how they may interfere with karyopherin mediated transport an/or chaperoning of RNA binding proteins. Nanaura et al use a series of in vitro assays to evaluate whether the poly(PR) DPR protein disrupts the association between KapB2 and the RNA binding protein FUS. While the manuscript presents interesting data from a protein interaction perspective, the findings are not particularly novel. Moreover, the implications for cellular pathophysiology in disease are completely unclear and would greatly strengthen the novelty and overall impact of this manuscript. In its current form, I can not recommend this paper for publication at Nature Communications. However, specific comments that if addressed would make this manuscript suitable for consideration at Nature Communications can be found below.

1. I find it rather odd that the authors focused on only PR. However, multiple DPR species have been shown to disrupt NCT (including a recent biorxiv paper regarding GA sequestration of karyopherins). Thus, the reason for selecting just PR remains unclear and additional DPR species would make for a nice comparison.

Thank you for your insightful assessment. As you mentioned, Frédéric Frottin et al. (2020) have revealed in a recent biorxiv paper that poly-GA aggregates sequester importins including importin α 1, α 3, and β 1 (ref.1). We performed additional turbidity assay with five different poly-dipeptides and revealed that PR/GR poly-dipeptides inhibited the chaperone activity of Kap β 2 (Fig.1c). These results correspond with recent studies in which arginine-rich poly-dipeptides were found to interfere with nuclear import receptors (ref. 2, 3). These results contributed to our decision to focus on the toxicity of arginine rich poly-dipeptides towards chaperone function of Kap β 2 in this study.

References:

1. Frottin F, Pérez-Berlanga M, Hartl FU, Hipp MS. Multiple pathways of toxicity induced by C9orf72 dipeptide repeat aggregates and G4C2 RNA in a cellular model. *bioRxiv*, 2020.2009.2014.297036 (2020).
2. Hayes LR, Duan L, Bowen K, Kalab P, Rothstein JD. C9orf72 arginine-rich dipeptide repeat proteins disrupt karyopherin-mediated nuclear import. *eLife* 9, (2020).
3. Hutten S, et al. Nuclear Import Receptors Directly Bind to Arginine-Rich Dipeptide Repeat Proteins and Suppress Their Pathological Interactions. *Cell Rep* 33, 108538 (2020).

2. It is also unclear why the authors focus on FUS. Is FUS pathology a prominent feature of C9 ALS/FTD? The in vitro binding experiments suggest that PR can associate with multiple karyopherins, including those that may be implicated in the transport and/or chaperoning of TDP-43 which is a prominent pathological feature of C9 ALS/FTD. Thus, the authors should additionally investigate this to increase relevance to disease pathophysiology.

Thank you for your advice. We performed additional hydrogel binding assay and observed that PR poly-dipeptides inhibited the ability of Imp α/β 1 in modifying phase transition of TDP43 (page 3, line 28-30, Supplementary Fig. 1e, f). As you mentioned, cytoplasmic aggregations of FUS are observed less frequently than those of TDP43 in post-mortem examinations, even though aberrant cytoplasmic localization of FUS and TDP43 are observed in C9-ALS/FTD pathology (ref. 4, 5), which might be derived from the difference in tendency for aggregation between FUS and TDP43. We have included a discussion of this issue in our updated manuscript (page 6, line 10-24).

References:

4. Keller BA, Volkening K, Droppelmann CA, Ang LC, Rademakers R, Strong MJ. Co-aggregation of RNA binding proteins in ALS spinal motor neurons: evidence of a common pathogenic mechanism. *Acta Neuropathol* 124, 733-747 (2012).
5. Boeynaems S, Bogaert E, Van Damme P, Van Den Bosch L. Inside out: the role of nucleocytoplasmic transport in ALS and FTL. *Acta Neuropathol* 132, 159-173 (2016).

3. My largest and most important concern is that this entire manuscript is based on in vitro data. Given the conflicting evidence in the literature regarding the effects of DPRs on NCT, it is essential that the authors demonstrate relevance of their findings to a living cellular system. While in a test tube PR may disrupt phase transitioning of FUS via KapB2 interference, does this phenomenon actually occur in a real cellular environment in the context of multiple karyopherins and NCT factors, which are often present in excess quantities.

As recommended, we performed additional immunoprecipitation (Fig. 2a, Supplementary Fig. 2a, b) and confirmed the interaction between Kap β 2 and PR poly-dipeptides in a cellular environment (Fig. 2a). Our results appear to correspond with the findings of a recent study (ref. 3). Though NIRs are abundant in a cellular environment, increased amount of NIRs suppress the pathological protein interaction by arginine-rich poly-dipeptides (ref. 3). A previous study also revealed that overexpression of KAP104, the yeast homolog of Kap β 2, strongly suppressed the toxicity of PR poly-dipeptides (ref. 6). Based on the evidence presented in these references, the relation between the quantities of NIRs and those of PR-polydipeptides might be crucial in a cellular environment. We have discussed this in further depth within our discussion section (page 6, line 40-45, page 7, line 1-6).

References:

3. Hutten S, et al. Nuclear Import Receptors Directly Bind to Arginine-Rich Dipeptide Repeat Proteins and Suppress Their Pathological Interactions. *Cell Rep* 33, 108538 (2020).
6. Jovicic A, et al. Modifiers of C9orf72 dipeptide repeat toxicity connect nucleocytoplasmic transport defects to FTD/ALS. *Nat Neurosci* 18, 1226-1229 (2015).

Minor Comments

1. Manuscript is hard to follow in places due to the fact that some sentences appear to be incomplete.

Thank you for bringing our attention to this. We have since rewritten the manuscript and sent it to a native English speaker for proofreading.

2. Stats are missing from multiple graphs.

Thank you for identifying this issue. We have performed additional statistical analyses and included our results in Figure 1f-i.

Reviewer #3:

In this study, together with experiments, the authors used molecular dynamics (MD) simulations to further prove the effective binding of proline:arginine poly-dipeptides in the predicted binding site.

I will only comment on the MD part. In this study, MD simulations are used in order to sustain the finding that poly-dipeptides can directly bind to the predicted binding site. Certainly, MD is a well-suited technique to study problems like the one tackled in this study. However, these simulations last 15ns, which is far too short nowadays. Any potential unbinding event would not be observable in such a short run. These simulations do not meet the current standards of MD simulations (typically hundreds of ns, at least, for a problem and model system like the one contained in this study). Indeed, a reasonable solution would be to extend the simulations to reach at least about 100ns). Moreover, the authors should show at least the root mean square deviation (RMSD) of both the protein and the poly-dipeptide, distinctly, during the simulations. This would reinforce the data and somehow sustain a bit more their findings.

We appreciate your constructive comments. As recommended, we performed additional 100ns MD simulation. We also demonstrated RMSD in our Supplementary Fig 6. Accordingly, we have rewritten our result and methods sections (page 5, line 2-7, page 11, line 37-44, page 12, line 1-11).

The authors should specify:

- the force fields used to model the protein, the poly-dipeptides as well as the sodium ions;

Thank you for your suggestion. We have rewritten our methods section on MD simulation (page 11, line 37-44, page 12, line 1-11).

- How the electrostatic potential was calculated on the protein surface.

Thank you for your advice. Surface electrostatic potential was calculated using Adaptive Poisson-Boltzmann Solver (version 2.1). The structure was drawn with PyMol program (version 2.3.0.) (page 12, line 10-11).

In summary, there is very little here that can be said in favor of such MD simulations. Details on the model system and parameteres are missing. The analysis of (only) 15ns is poor, and my sense is that MD here is very marginal in the overall story.

As suggested, we performed additional 100ns MD simulation and observed that the results of MD simulation were highly consistent with data from NMR (page 5, line 2-7). Biochemical experiments were corroborated by MD simulation; therefore, we think MD simulation strengthened our message that PR poly-dipeptides partially bind to the NLS binding site of Kap β 2.

REVIEWERS' COMMENTS

Reviewer #1 (Remarks to the Author):

The current version of the manuscript from Nanaura et al. is much improved by the work the authors have put in. I appreciate the additional experiments to characterize the relative affinities of PR dipeptides and M9 NLS-containing proteins. Further, I believe the current discussion does a much better job of placing the results of this work in the greater context of dipeptide repeat pathology and provides some very interesting speculation about the implications of these findings as well. All of my previous concerns have been addressed.

I have several minor points/suggestions:

1. Perhaps use one sentence to better explain what a low-complexity sequence is.
2. "Phase transition" is used in several places. Instead use "a phase transition"
3. Line 41: adopter should be adaptor
4. I would include this line from the methods in the main text so that the readers have a better idea of what type of MD simulation is being discussed "The input structure for MD simulations was prepared by taking chain A of the crystallographic dimer from the crystal structure of Kap β 2 in complex with FUS (PDB: 5YVG) and mutating side chains of chain X to PR."
5. It might be useful, perhaps in figure 5, to include a structure of KapB2 showing csps from the M9 peptide and poly PR to visualize the overlapping binding pockets as defined by NMR.
6. The methods should include the parameters used in APBS to calculate the surface potential. Specifically, what ionic strength was used.

Reviewer #2 (Remarks to the Author):

The authors have revised the manuscript suitably and the new data answers my earlier concerns

Reviewer #3 (Remarks to the Author):

The authors have tried to address my comments, and the MD part has improved. The MD simulations are now 100 ns (vs 15 ns before), which remains very limited. The mechanistic insights from MD are very marginal, in this story. MD is ancillary, while this tool can be quite informative, and that's why I remain a bit disappointed with its use in this investigation. Nevertheless, the overall paper is interesting, and now the MD part is okay, being somehow accessory to the experimental evidence.

Point-by point response to reviewers' comments

Reviewer #1:

The current version of the manuscript from Nanaura et al. is much improved by the work the authors have put in. I appreciate the additional experiments to characterize the relative affinities of PR dipeptides and M9 NLS-containing proteins. Further, I believe the current discussion does a much better job of placing the results of this work in the greater context of dipeptide repeat pathology and provides some very interesting speculation about the implications of these findings as well. All of my previous concerns have been addressed.

I have several minor points/suggestions:

- 1. Perhaps use one sentence to better explain what a low-complexity sequence is.*
- 2. "Phase transition" is used in several places. Instead use "a phase transition"*
- 3. Line 41: adopter should be adaptor*
- 4. I would include this line from the methods in the main text so that the readers have a better idea of what type of MD simulation is being discussed "The input structure for MD simulations was prepared by taking chain A of the crystallographic dimer from the crystal structure of Kap β 2 in complex with FUS (PDB: 5YVG) and mutating side chains of chain X to PR."*
- 5. It might be useful, perhaps in figure 5, to include a structure of KapB2 showing csps from the M9 peptide and poly PR to visualize the overlapping binding pockets as defined by NMR.*
- 6. The methods should include the parameters used in APBS to calculate the surface potential. Specifically, what ionic strength was used.*

We appreciate your suggestions. As you recommended, we have revised the manuscripts as follows:

1. We added one phrase to explain what a low-complexity sequence is (page 2, line 15-17).
2. We rewrote "phase transition" to "a phase transition/phase transitions".
3. We modified an error in spelling.
4. We moved the highlighted sentence from the methods section to the main text (page 5, line 9-11).
5. We added a new panel in Fig. 4k to make it easy to refer to the residues which seems to interact with M9M and PR poly-dipeptides.
6. We added the following sentence to explain which parameters we used in calculating the surface potential (page 12, line 17-19): "Electrostatic potential of Kap β 2 was calculated by PyMOL APBS tool at 0.15 M of monovalent salt (± 1) and represented with ± 5 kT/e range."

We thank you again for taking time to provide constructive comments on our manuscript.

Reviewer #2:

The authors have revised the manuscript suitably and the new data answers my earlier concerns.

We thank you again for your reviewing the manuscript.

Reviewer #3:

The authors have tried to address my comments, and the MD part has improved. The MD simulations are now 100 ns (vs 15 ns before), which remains very limited. The mechanistic insights from MD are very marginal, in this story. MD is ancillary, while this tool can be quite informative, and that's why I remain a bit disappointed with its use in this investigation. Nevertheless, the overall paper is interesting, and now the MD part is okay, being somehow accessory to the experimental evidence.

We appreciate your comments to the manuscript.